# Cervical Cancer Genetic Profile through Circulating Tumor DNA: What Can We Learn from Blood?

**DOI:** 10.3390/biom14070825

**Published:** 2024-07-10

**Authors:** Sevastiani Antonouli, Valentina Di Nisio, Nikoletta Daponte, Athina-Ioanna Daponte, Alexandros Daponte

**Affiliations:** 1Department of Obstetrics and Gynaecology, Faculty of Medicine, School of Health Sciences, University of Thessaly, 41500 Larisa, Greece; arella_935@hotmail.com (S.A.); nikolettadaponte@gmail.com (N.D.); 2Department of Gynecology and Reproductive Medicine, Karolinska University Hospital, Huddinge, 141 86 Stockholm, Sweden; valentina.di.nisio@ki.se; 3Division of Obstetrics and Gynecology, Department of Clinical Science, Intervention and Technology, Karolinska Institutet, Huddinge, 171 77 Stockholm, Sweden; 4Second Department of Dermatology-Venereology, Aristotle University School of Medicine, 54124 Thessaloniki, Greece; atdaponte@uth.gr

**Keywords:** cervical cancer, HPV, liquid biopsy, ctDNA

## Abstract

Cervical cancer (CC) is one of the deadliest gynecological cancers worldwide. Human papillomavirus is the main etiological agent responsible for the initiation and development of most CC cases. The standard method utilized for CC screening in the global population is the cytological Pap smear test. Despite its effective validity in detecting precancerous lesions and its response to layer stages of this disease, greater screening and diagnostic reliability are needed, as well as an improvement in specificity and sensitivity. In this context, the use of liquid biopsies, like blood, for the isolation of circulating tumor DNA (ctDNA) in CC screening, diagnosis, prognosis, and surveillance could fill the gaps that still exist. In the present review, we aim to study the literature in order to collect knowledge on blood-based liquid biopsy based on descriptions of its precious molecular content and its utilization as a potential tool for CC patients’ management. We will mainly focus on the important role of the novel ctDNA and the unique possibilities to additionally use HPV-ctDNA in CC at various stages of clinical application.

## 1. Introduction

Cervical cancer (CC) is defined as the malignant transformation of cells in the uterine cervix. The classification of the World Health Organization includes the premalignant lesions as cervical intraepithelial neoplasia (CIN), and divides histological epithelial tumors into squamous cell carcinoma, glandular, mesenchymal, mixed epithelial and mesenchymal, melanocytic, miscellaneous, lymphoid and hematopoietic, and secondary tumors [1,2]. The oncogenesis of this type of gynecological cancer is mostly correlated with infection from human papillomavirus (HPV), with at least 80% of cervical adenocarcinoma associated with HPV [3], alongside smoking, early sexual initiation, multiple sexual partners, and long-term oral contraceptive consumption [4]. HPV is an infectious carcinogenic agent that invades the epithelium of the cervix, oropharynx, and anus, integrating its DNA with the host cells and triggering the tumorigenic onset [5]. In CC, HPV comprises more than 200 identified genotypes that can be classified as “high-risk” (HR)-HPV (e.g., HPV16, 18, and 39), and benign lesion-responsible “low-risk” (LR)-HPV (e.g., HPV11, 44, and 61) [6]. Among the HR-HPVs, HPV16 (mainly in squamous cell carcinoma) and HPV18 (mainly in adenocarcinoma and adenosquamous carcinoma) account for 70% of total CC cases, and are thus prioritized in screening [7,8,9,10,11,12,13] and targeted by the bivalent vaccine [14,15]. Abnormal cells caused by HPV have usually been detected at the time of Pap smear during CC screening, even if this method is not applicable for potential recurrence estimation [16]. In addition, the so-called “molecular” Pap test (HPV-DNA test), different from the cytological one, is considered the golden standard for primary screening [17,18], as reported in the guidelines from the American Cancer Society [19] and the American College of Obstetricians and Gynecologists [20], and the European Guidelines for Quality Assurance in Cervical Cancer Screening [21].

The disease stage and metastasization in lymph nodes are the main CC prognostic factors at diagnosis [22,23], together with the tumor size [24,25]. The early CC stages (up to FIGO 2018 stage IB2-IIA1) can be treated surgically by a radical hysterectomy with bilateral pelvic lymphadenectomy, showing a low two-year recurrence risk (ca. 8–10%) [22], while chemoradiotherapy is required for advanced stages (FIGO 2018 stage IB3-IVB), with a 30–50% two-year relapse risk [26,27]. Despite regular clinical examination via magnetic resonance imaging or computer tomography, it is still challenging to predict in which patients CC will potentially recur after treatment. The standard of care for identification and follow-up of CC has relied on tissue biopsies, which are invasive and accurate only in the retrieved part and thus difficult to generalize from as a general cancer-stage or treatment-response evaluation, especially considering the factor of tumor heterogeneity [28,29]. To overcome these limitations, the use of liquid biopsies (LB), such as blood, urine, saliva, or any other bodily fluid, could be a more patient-friendly, non-invasive, and illuminating method. Among other biological components, LBs contain cell-free DNA (cfDNA), particularly circulating tumor DNA (ctDNA) and ctDNA originating from HPV (HPV-ctDNA) that could be used as a powerful analyte for cancer patients’ management [30,31,32] (Figure 1). In fact, specific DNA-related modifications such as oncogenes (e.g., viral E6 and E7), tumor-suppressor mutations (e.g., *TP53* and *RB*), microsatellite alterations, and epigenetic changes such as hypermethylation can be detected [33]. However, studies report limited success in the detection of serum and plasma HPV-ctDNA, pointing to the necessity of developing more suitable technologies in DNA isolation and detection [34,35,36]. The relevance of LB, with particular focus on ctDNA and HPV-ctDNA potential as CC biomarkers, will be discussed in the following sections of the present review.

## 2. Blood-Based Protein Biomarkers in Monitoring Cervical Cancer and the Novel Circulating Tumor DNA

The new technologies applied in the monitoring of malignant neoplasm rely on the measurement of biomarkers from easily accessible and non-invasive samples, such as serum. Cervical cancer (CC) can be divided in two main histological subtypes, namely squamous cell carcinoma (>80% of the cases) and adenocarcinoma (ca. 12% of the cases) [37]. Among the most used protein-based biomarkers, the squamous cell carcinoma antigen (SCC-Ag) and cancer antigen 125 (CA-125) are the most reliable for CC staging, evaluating the status of lymph nodes and the clinical outcome [38,39]. Additionally, fragments of cytokeratin (CYFRA), carcinoma embryonic antigen (CEA), immunosuppressive acidic protein (IAP), cyclooxygenase-2 (COX-2), soluble CD44, matrix metalloproteinase (MMPs), and their tissue inhibitors (TIMPs) are used as diagnostic and prognostic protein biomarkers, even though their levels fluctuate considerably depending on the cancer type and stage [40]. The lack of prospective randomized trials impairs the advancement of these biomarkers as standards for patient outcome improvement. The main reason could lie in the low cancer specificity and low sensitivity of the aforementioned biomarkers for low-volume cancer recurrence.

The landscape of biomolecules detectable in serum LB is enriched by molecular biomarkers, such as the high-risk human papilloma virus (HPV) DNA [41]. Alongside cervical cytology, the detection of serum HPV-DNA can be used in CC diagnosis as it can also detect the presence of an oncological case in negative cytology test results [42,43].

The novel approach in CC screening and potential diagnostic and/or prognostic markers is the detection and measurement of biomolecules present in LBs (terminology that includes both blood and other bodily fluids) (Figure 1). In this frame, circulating tumor cells, exosomes with DNA and/or RNA cargo, and extracellular DNA strands that therefore take the name of cell-free DNA (cfDNA; ca. 166 bp) constitute a powerful tool in the detection and follow-up of cancer patients’ health [44]. On average, 6 ng of cfDNA can be isolated for each mL of plasma, of which a very low percentage or none at all is derived from cancer cells [45]. Nevertheless, circulating tumor DNA (ctDNA) is the tumor-released cfDNA that can be more easily detected in LBs, through ultra-sensitive targeted methods (e.g., digital PCR) and next-generation sequencing (NGS) technology, and has the potential to be used as a valuable diagnostic/prognostic biomarker [46]. The versatility and potential of ctDNA resides in its close resemblance in genetic and epigenetic modifications to the DNA of the generating cells, and in the accuracy as to the current status of cancer cells due to its very short half-life (~2.5 h) [47,48,49]. Additionally, its levels in plasma rise in advanced cancer stages (ca. 30 ng/mL at stage III), up to 1000 ng/mL in cases of metastasis [50,51], depending on the tumor’s size and its metabolic burden [52,53,54]. In a broader perspective, the kinetics of ctDNA release have not yet been fully comprehended, considering the low median variant allele frequency detected (<4 mutated alleles out of 1000 ctDNA copies) [55]. For this reason, gene amplification of more than five copies can be easily detectable compared to duplication, and even more so to single mutation [56].

The use of ctDNA as a cancer diagnostic, prognostic, postoperative surveillance, and close therapy follow-up biomarker has multiple advantages compared to the protein-based strategy. Several characteristics of this new biomarker have been applied in the current oncological field for estimating the tumor load and cancer outcome [57] and designing personalized therapies [58], as well as monitoring residual disease/relapse [59] and performing population screening [60]. For instance, ctDNA concentration, chromosome modification, and gene methylation are the most straightforward [61], even though their identification still lacks the proper sensitivity and specificity to be applied as a standard of care in CC cases. Additionally, a great advantage in the use of ctDNA as a biomarker is its unique potential for the detection of circulating viral DNA copies [46,62]. Its first application to CC was recorded in 2012, when Campitelli et al. found that HPV insertion could be detected in ctDNA of patients with CC over stage I [63].

## 3. The Potential Tumor ctDNA and HPV-ctDNA Blood-Based Biomarkers in Cervical Cancer

Nowadays, ctDNA detection and analysis are considered useful tools in the diagnosis, prognosis, and treatment guidance of gynecological tumors, including CC [64]. DNA fragments can be of both human and viral origin, offering pivotal information for monitoring CC patients [65]. Until now, several efforts in CC have been made to identify cancer-related genes and their clinical validity through tumor-tissue biopsy sequencing analysis. PIK3CA, TP53, KRAS, STK1, and PTEN are some of those cancer-related genes proven to either carry isolated somatic mutations or copy number alterations, including in CC cases [66,67,68,69,70]. Other genetic alterations to be identified in CC include MAPK1, EGFR, HLA-B, EP300, FBXW7, NFE2L2, and ERBB2 in squamous cell carcinoma and ELF3 as well as CBFB somatic mutations in adenocarcinoma [71,72]. A recent study performing an extensive multigene NGS with a panel targeting 571 CC-related genes identified a total of 810 somatic variants, 2730 germline mutations, and 701 copy number variations (CNVs), with FAT1, HLA-B, PIK3CA, MTOR, KMT2D, and ZFHX3 shown to be the most mutated genes and PIK3CA, BRCA1, BRCA2, ATM, and TP53 gene loci to have a higher frequency of CNVs [73]. Notably, a shorter overall survival (OS) has been evidenced in patients of early or advanced CC with mutant KRAS and PIK3CA genes [66,71,74], as well in patients with squamous cell carcinomas and EGFR amplification [75]. However, testing specificity and sensitivity are still controversial in CC detection, and more trials are needed to establish this molecular identification.

Even if CC tumor-tissue biopsy is an efficient tool in detecting genetic alterations and has held clinical significance for many years, over the past decade, ctDNA LB has gained great attention due to its potential as a minimally invasive biomarker in the carcinogenic process. This novel blood-based biomarker might provide information for monitoring CC by reflecting the tumor load according to ctDNA concentration, DNA fragment size, and epigenetic alterations such as methylation patterns and chromosome changes [61], even though the amount of ctDNA in the bloodstream depends on several factors and is representative of the total plasma cfDNA. Additionally, in viral-related cancers such as CC, the non-human origin of the viral DNA enables the use of circulating HPV-DNA as a specific and promising biomarker in liquid-biopsy-based assays [76], especially for CC diagnosis [77,78], prognosis [79], and early relapse or recurrence prediction [80,81], though not yet for early CC detection [82,83]. To our knowledge, integration of the HPV genome into cervical cells typically results in an increased expression and stability of transcripts encoding the viral oncogenes E6 and E7, simultaneously favoring common fragile sites and different chromosomal loci, thus showing a mutagenesis pattern in the APOBEC cytidine deaminase alongside a high degree of chromosomal instability [84]. It is found that HPV-ctDNA E7 fraction could be a potential tumor marker, highly specific and moderately sensitive for detection and monitoring of minimally invasive cervical cancer [85]. Among all approaches of detecting HPV circulating DNA in CC blood samples, ddPCR has proved superior due to its accuracy and sensitivity [33]. Similarly, through ddPCR application in blood samples to detect HPV circulating DNA, CC patients with increased viral load have been linked to a five-year high risk of recurrence and low survival rate [86]. Four years ago, a trial for HPV-DNA genotyping associated with CC development employed an NGS method based on targeted amplicon for early and precise detection of high-risk HPV genotypes developed with relevant sensitivity [87]. Recently, some authors, by calculating the ratio of HPV-ctDNA in 64 plasma samples of CC patients, suggested that the HPV-ctDNA/total-ctDNA ratio after targeted NGS may be a biomarker for treatment responses, monitoring, and prediction [88]. HPV-DNA methylation was also studied by a three-marker panel in urine and plasma ctDNA of CC patients, showing in both liquid ctDNA biopsy samplings a high sensitivity, and specificity in the detection of more advanced CIN2+ (i.e., CIN2 and CIN3) lesions that can progress towards CC [89]. Indeed, in patients with genotype-specific HPV-associated carcinoma, the blood sample could be instrumental for a complete viral genome characterization for diagnostic purposes [78]. Blood-based LB ctDNA detection is aimed at meeting requirements for HPV-ctDNA identification and cancer monitoring and follow-up. Nevertheless, to date, there is not sufficient proof to clinically validate the use of ctDNA in CC treatment, only some evidence proposing this novel biomarker as a complementary diagnostic tool [90].

Recent investigations of CC dealing with the amount of ctDNA in plasma and the connection with clinicopathological characteristics showed that the ctDNA content of CC was much higher than that of the control group and significantly different in histological grade, infiltration depth, lymphatic metastasis, and Federation of Gynecology and Obstetrics (FIGO) stage [91]. However, published literature is still focusing on mutation detection in CC [46,90,92], either through targeted NGS panels or ddPCR to detect ctDNA and identify gene variants that may be useful for diagnosis, prognosis, and MRD and tumor monitoring, as well as for guiding therapy.

There have been a few studies until now revealing the current mutation profile of CC cancer through ctDNA blood-based analysis. Plasma CADM1 methylation is a promising metastasis marker in CC cancer [93]. In ctDNA from blood, PIK3CA, ZFHX3, KMT2C, and KMT2D were shown to be the most frequent mutated genes, yet not affecting all studied patients [94]. In 2017, Chung et al. reported two single PIK3CA mutations, p.E542K and p.E545K, when analyzing ctDNA on plasma samples by ddPCR, and identified them in 22.2% of Chinese patients with primary invasive CC. These alterations of the PIK3CA gene correlated mainly with decreased disease-free survival and increased tumor size [95]. The next year, in 2019, when searching for copy number alterations in pre-surgery plasma samples from 11 CC patients, Nakabayashi et al. detected them in 3 stage I–II patients, with decreased progression-free survival (PFS) and OS rates [96]. Concomitantly, in 2019, Tian et al. used an NGS targeted exome sequencing in 48 tumor-related genes on isolated ctDNA from 93 plasma samples of 57 Chinese CC patients to identify pathogenic mutant genes and their frequency. The KDR, ALK, EGFR, and ATM mutant genes had high frequency, whereas PDGFRA, CSF1R, ERBB2, HNF1A, NOTCH1, TP53, APC, KRAS, and PTPN11 had lower frequency. In addition, by developing an algorithm that gives the allele fraction deviation (AFD), measured through mutant allele fraction detected in ctDNA through germline mutation analysis of each patient, the researchers found a decrease in AFD value in CC patients following treatment, together with reduced tumor size. A high AFD value was associated with disease progression and the initiation of metastasis, and low baseline AFD in diagnosis followed by an increase later on was linked to relapse [97]. Also in 2019, in a small study of four patients with CC, among other gynecological tumors, by using a CAPP-Seq-based NGS approach with a 197-gene panel, Iwahashi et al. detected non-synonymous mutations in all patients with PIK3CA and EGFR mutant genes, in both tumor and plasma samples [98].

Along with the other research groups in 2019, Lee et al. employed an NGS customized panel to characterize single-nucleotide variants (SNVs), insertion–deletion polymorphisms (INDELs), and CNVs in 51 candidate genes and performed an analysis of isolated ctDNA and circulating tumor-cell DNA (ctcDNA) from blood samples of four CC patients, among other samples of patients with different gynecological cancers. Common mutated genes were RCA2, ERBB2, ESR1, FGFR4, PTCH1, STK11, and TSC2, and the data relating the variant detection and mutated genes in all patients showed nine common genetic variants between ctDNA and ctcDNA sequencing analysis, thus contributing to a better understanding of tumor heterogeneity and the development of new targeted therapies [99]. One year later, in 2020, the same group conducted a study in 24 CC patients before primary treatment by using an NGS customized panel with fewer genes than before. A 24-gene panel was designed and mutations were found in 18 genes, with somatic alterations in ZFHX3, KMT2C, KMT2D, NSD1, ATM, and RNF213, with the lowest frequency of 27% in RNF213 and the highest of 83% in ZFHX3 genes. Their data revealed also that RNF213 mutation could be a valuable marker to monitor further chemotherapy and radiotherapy treatments [100].

More recently, in 2021, Charo et al. evaluated the sequencing results of isolated ctDNA from 13 CC patients, among other gynecological cancer patients, with the use of a 73-gene targeted NGS panel in a PREDICT clinical trial (NCT02478931). In these CC patients, five common gene mutations were identified—PIK3CA, TP53, FBXW7, ERBB2, and PTEN—with PIK3CA the most present, followed by the others accordingly. They showed that higher ctDNA maximum mutation allele frequency was connected with worse OS [hazard ratio (HR): 1.91, *p* = 0.03], while therapy matched to ctDNA alterations was independently associated with improved OS (HR: 0.34, *p* = 0.007) compared to unmatched therapy in multivariate analysis, thus validating the higher mutation allele frequency as a possible clinical predictive and prognostic factor for OS in such patients [101]. The same year, in a Chinese cohort of 10,000 patients, Zhang et al. performed a large ctDNA study, including 123 plasma samples from patients with CC, among other cancers, and deep NGS sequencing of 1021 genes. The analysis elucidated mutated genes PIK3CA, MLL3, TP53, MLL2, EP300, PTEN, FGFR3, DNMT3A, PTCH1, and TERT, with PIK3CA the most frequent (30%), and the CH-related mutations in DNA methyltransferase 3 alpha (DNMT3A) the most frequent (5.9%) in CC patients. Including CC patients, the sensitivity of ctDNA detection in stage I–III disease was >60%, and in stage IV >70%, thus allowing the conclusion that such ctDNA findings in a large cohort may contribute to potential new combined treatment strategies and targeted personalized therapies [102]. In 2021, another group of researchers, Tian et al., performed a deep sequencing analysis on 322 cancer-related genes using plasma samples of 82 CC patients either locally advanced or metastatic-relapsed. In the isolated ctDNA from plasma, five specific nonsynonymous mutated genes were identified in the metastatic patient cohort (PIK3CA, BRAF, GNA11, FBXW7, and CDH1) and associated with a significantly shorter PFS (HR 2.57, 95% CI: 1.20–5.52, *p* = 0.005) and OS (HR 2.66, 95% CI: 1.20–5.87, *p* = 0.007). Interestingly, they also demonstrated that continuous monitoring with ctDNA in plasma samples (23 metastatic CC patients underwent serial ctDNA analysis) can offer valuable predictive and prognostic information to guide decision-making in CC patients, as their data showed an increase of such mutations prior to disease progression response to radiology treatments [103].

In conclusion, large steps have been taken in the last several years regarding the LB non-invasive approach in terms of using ctDNA analysis in CC patients’ plasma samples and ctDNA-HPV analysis. The combination of genetic markers (ctDNA) with circulating HPV could be incorporated in a CC-specific panel of biomarkers to aid in the prediction of prognosis and patient-specific targeted therapy. Even if still few, these studies conducted in plasma ctDNA can provide preliminary data in gene-mutation profiling of CC patients that are predominantly indicative either of disease progression or survival. The increased sensitivity gained by applying such methods in CC should be taken under consideration, together with ultra-deep sequencing approaches, in order to provide a large gamma of data, so as to enable plasma ctDNA use in routine clinic application when managing CC patients.

## 4. What Is Still Needed for ctDNA to Become a Valuable Tool in Routine Clinical Diagnostics of Cervical Cancer?

In a broader perspective, ctDNA contribution in the clinical routine is gaining more ground, even though most of the research is still at a proof-of-concept level. The assessment of this novel powerful diagnostic/prognostic biomarker has been approved by the U.S. Food and Drug Administration (FDA) for a limited range of cancers, including breast, prostate, and non-small-cell lung cancer [104,105]. Among those with pre-market approval, Agilent Resolution ctDx FIRST assay [106], Myriad’s BRACAnalysis CDx^®^ [107], Epi proColon^®^ [108], Roche’s cobas^®^ EGFR Mutation Test v2 [109], FoundationOne^®^Liquid CDx [110], and Guardant360^®^ CDx [111] are included, covering applications such as early detection, screening, gene-targeting, relapse detection, measurable residual disease, and more (extensively reviewed in [112]). Narrowing down the topic to HPV-ctDNA, recently the Naveris NavDx^®^ test [113] has been clinically validated for the detection of ctDNA in HPV-positive cancers, contributing to the optimization of patients’ clinical management, although it is not yet approved by the FDA. In addition, clinical trials are ongoing for the detection of HPV-ctDNA and its association with treatment response and its potential as a biomarker for surveillance and disease recurrence (NCT04574635, NCT03739775, [114,115]), as well as for screening purposes (NCT04857528, NCT04274465).

The quest for a standardized clinical application of ctDNA in CC during the diverse phases of diagnosis, prognosis, and follow-up is dotted with several challenges. The first technical drawback is the generation of false-negative or -positive results due to the possible discordance of tumor tissue and serum/plasma mutated ctDNA content. In addition to the technical challenge of collecting the correct amount (ca. 4–5 mL) of serum/plasma with enough ctDNA copied for the analysis, we need to underscore that the mutation could be clonal or subclonal, increasing the difficulties of detection of variants of low allele frequency [116]. This challenge becomes even more marked considering that ctDNA content is highly variable depending on tumor type and on patient-specific level [51]. From a different technical perspective, the ongoing improvement of DNA library preparations and ultra-deep sequencing methods could be a solution to false-negatives [117,118,119,120,121], taking under consideration also the size of targeted ctDNA fragments [121]. On the other hand, the presence of multiple mutations could give false-positive results due to errors during DNA library preparation and poorly followed procedures. To overcome this challenge, the use of a molecular barcode during NGS analysis or a reliable bioinformatic pipeline for downstream data analysis could be used, in combination with a paired comparison of tumor-serum/plasma of variants bordering the limit of detection and the removal of somatic mutation identified through white-blood-cell parallel analysis [116,118,122,123,124,125,126].

In the CC context, the HPV-ctDNA isolation and detection could be coupled with the standard Pap smear test when a large amount of evidence exists and may support its accuracy and validity, giving a chance to provide insights on the lesion stage and a better follow-up process. Furthermore, the probable use of non-invasive blood samples as LB for isolating HPV-ctDNA may also offer a possibility to expand the screening to developing countries, even though this would come with some limitations. Additionally, as mentioned before, while early-stage CC could result in a false-negative due to the low fractions of viral ctDNA [79,82], this method would be extremely helpful in detecting locally advanced CC [79,81,82].

## 5. Conclusions

Overall, the novel world of LB is opening new avenues for medical health care and prevention in oncological patients. The minimally invasive, rapid, and low-cost sampling could be an additional benefit to the actual content: tumor biomarkers comprising ctDNA from both the tumor mass and the viral agent in HPV-positive cancers. Since ctDNA testing is still in the research stage, and there is a lack of standardized management, the effectiveness and practicality of ctDNA testing still needs further exploration before its application in clinical practice. Nevertheless, the current studies and clinical trials reported in the present review highlight the recent advancement in the field and the promise that ctDNA could be instrumental for personalized therapy and precision medicine, including in CC patients.

## Figures and Tables

**Figure 1 biomolecules-14-00825-f001:**
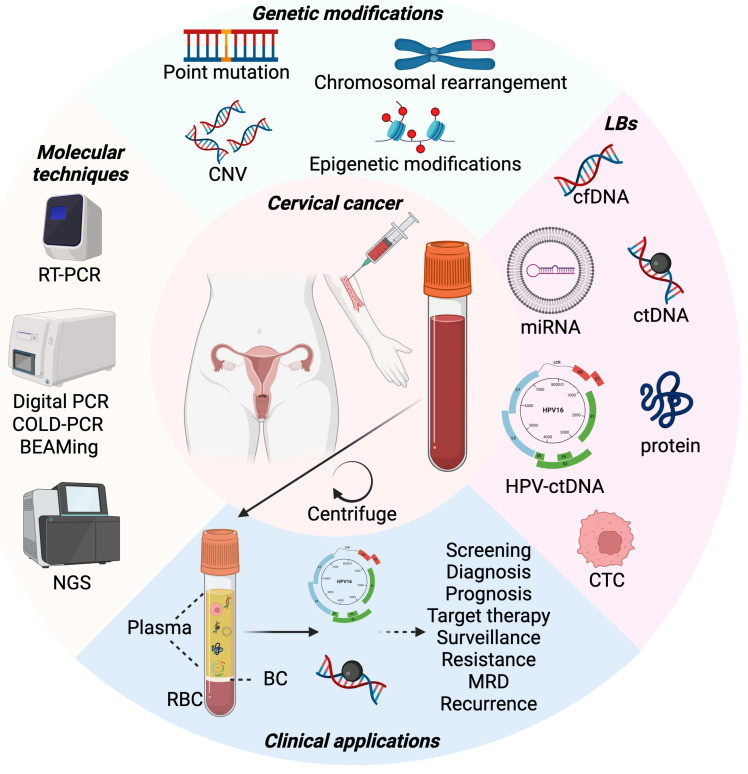
Overview of blood-based ctDNA sampling analysis: genetic alterations, molecular techniques, clinical applications. By the means of blood sample as LB, several types of markers—such as CTC, cfDNA, ctDNA, viral HPV-ctDNA, proteins, and miRNAs carried in extracellular vesicles—can be detected in plasma of CC-affected patients. Through several molecular techniques, including RT-PCR, digital, COLD-, and BEAMing PCR, and NGS, genetic alterations on the mutation status of chromosomes, DNA, and epigenetic landmarks can be detected. LB-derived material can be instrumental for several clinical applications, including screening, diagnosis, prognosis, surveillance, resistance, MRD, and recurrence of CC, in addition to guiding the choice of targeted therapies for maximizing the outcome. Abbreviations: BC, buffy coat; BEAMing, beads, emulsion, amplification, magnetics PCR; CC, cervical cancer; cfDNA, cell-free DNA; CNV, copy number variation; COLD-PCR, co-amplification at lower denaturation temperature- polymerase chain reaction; CTC, circulating tumor cell; ctDNA, circulating tumor DNA; HPV, human papilloma virus; MRD, minimal residual disease; NGS, next-generation sequencing; RBC, red blood cells; RT-PCR, real-time reverse transcription–polymerase chain reaction. Created with BioRender.com.

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
