# Peer review of "Cervical Cancer Genetic Profile through Circulating Tumor DNA: What Can We Learn from Blood?"

_biomolecules, 2024, doi:10.3390/biom14070825_

Round 1

Reviewer 1 Report

Comments and Suggestions for Authors

In this work, the authors have summarized the potential use of blood-based liquid biopsy as a potential tool for CC screening and patients’ management. In general, this manuscript is well-written and informative. I have a few suggestions to the authors only:

1. Several tumor markers were mentioned (e.g. PIK3CA, TP53, KRAS, STK1, and PTEN). However, they are not CC specific. Please explain how they can be used in CC screening and the sensitivity and specificity of the test.

 2. If the figure is made by Biorender, it should be mentioned in the figure legend.

Author Response

In this work, the authors have summarized the potential use of blood-based liquid biopsy as a potential tool for CC screening and patients’ management. In general, this manuscript is well-written and informative.

Answer: Thank you for your valuable suggestions. Following your opinion at your first comment regarding the specificity of the tumor markers in CC, we add an explanation to clarify their use. Since indeed the figure is made by Biorender, we also mentioned it in the figure legend as you suggested.

I have a few suggestions to the authors only:

  1. Several tumor markers were mentioned (e.g. PIK3CA, TP53, KRAS, STK1, and PTEN). However, they are not CC specific. Please explain how they can be used in CC screening and the sensitivity and specificity of the test.

Answer: Thank you for your comment. We modified this part as suggested:

Lines168-170: “PIK3CA, TP53, KRAS, STK1, and PTEN are some of those cancer-related genes proved to either carry isolated somatic mutations or copy number alterations, including in CC cases”

Lines 179-181: “However, testing specificity and sensitivity are still controversial in CC detection, and more trials are needed to establish this molecular identification.”

  1. If the figure is made by Biorender, it should be mentioned in the figure legend.

Answer: Added.

Reviewer 2 Report

Comments and Suggestions for Authors

The manuscript “Cervical cancer genetic profile through circulating tumor DNA: What can we learn from blood?” is devoted to the important topic of more accurate diagnosis of cervical cancer, which will potentially help with earlier detection of relapses and serve to prevent them. The review describes the knowledge accumulated in the literature about tumor and viral DNA circulating in biological fluids (ctDNA), which could serve as biomarkers of precancerous or postoperative changes. The review can be published after a minor revision:

Abbreviations are not deciphered: ctcDNA, OS, PFS

Line 177: high degree of chromosomal - the phrase is unclear

Line 191: What are CIN2+ lesions? Please briefly explain

Line 294, typo: proxstate

Line 319 What is meant by “molecular barcode or a reliable bioinformatic pipeline”? Please clarify

Author Response

The manuscript “Cervical cancer genetic profile through circulating tumor DNA: What can we learn from blood?” is devoted to the important topic of more accurate diagnosis of cervical cancer, which will potentially help with earlier detection of relapses and serve to prevent them. The review describes the knowledge accumulated in the literature about tumor and viral DNA circulating in biological fluids (ctDNA), which could serve as biomarkers of precancerous or postoperative changes.

Answer: Thank you a lot for your helpful comments. Taking into consideration your opinion on our manuscript, we modified the text following all your comments and we clarify the suggested parts. Please, see the new lines corresponding to each of your comments:

Abbreviations are not deciphered: ctcDNA, OS, PFS

Answer: Full names added in line 258 (ctcDNA), line 277 (OS), line 295 (PFS)

Line 177: high degree of chromosomal - the phrase is unclear

Answer: Amended, new line 197: “high degree of chromosomal instability”

Line 191: What are CIN2+ lesions? Please briefly explain

Answer: Amended, new lines 211-212: “for the detection of more advanced CIN2+ (i.e., CIN2 and CIN3) lesions, that can progress towards CC”

Line 294, typo: proxstate

Answer: Amended.

Line 319 What is meant by “molecular barcode or a reliable bioinformatic pipeline”? Please clarify

Answer: Amended in lines 346-350: “To overpass this challenge, the use of molecular barcode during NGS analysis or a reliable bioinformatic pipeline for downstream data analysis could be used, in combination with paired comparison of tumor-serum/plasma of variants borderline to the limit of detection and the removal of somatic mutation identified through white blood cell parallel analysis”

Reviewer 3 Report

Comments and Suggestions for Authors

The article addresses an important topic concerning usability of circulating tumor DNA analysis in cervical cancer management. The amount of information in this area is growing rapidly, which makes the emergence of new reviews, involving the most recent data, certainly useful for the reader who wants to get more familiar with the topic.

However, I have a few comments on the conclusions that the authors make based on the reviewed data.

Some phrases sound very strange for me, i.e. ‘The standard method utilized for CC screening in the global population is the histological Pap smear test that gives a response on later stages of this disease, albeit missing the modification on precancerous lesions or post-therapeutic changes and metastatization’ (line 18). Did the authors mean cytological Pap smear? Even if so, cytological Pap smear, however subjective it could be, proved to be effective at detecting cervical precancerous lesions.

The statement ‘human papillomavirus (HPV) contributes in the 99% of the cervical cancer cases’ (line 36) is not supported by the large body of evidence. Even in the cited article of Kusakabe et al. it is stated that only approx. 95% of squamous cervical carcinomas are caused by HPV, and in cervical adenocarcinomas the proportion of HPV-negative cases is 10-15%.

The formulation ‘HPV detection has been routinely assessed by Pap smear during CC screening (line 46) looks confusing as the term ‘Pap’ relates to the mode of staining of the smear which is not used for HPV detection.

The definition of HPV-DNA test as “molecular” Pap test (line 48) is not common and, to my view, not correct. This term is much more convenient for the molecular gene expression profiling of the epithelial cells, reflecting the risk of cervical lesions, than for the HPV DNA testing.

Without a doubt, multiple biomarkers of cervical lesions can be of some value in patient management and successfully complement each other. However, in my opinion, the authors, commenting on literature data, substantially exaggerate the potential for detecting circulating HPV DNA using liquid biopsy. The end of the statement ‘Alongside cervical cytology, the detection of serum HPV-DNA can be used to detect the presence of an oncological case also in negative cytology test results, making it the only tool for proper CC diagnosis’ (lines 94-96) definitely must be softened.

The situation when as much as ‘4% of cfDNA in plasma is derived from cancer cells’ (lines 102-103) is not universal in cervical cancers, and even with invasive squamous cell carcinoma, and even more so with HSIL, HPV DNA cannot always be detected in plasma, that is supported by multiple publications in the field. In this regard, the negative predictive value of detecting circulating DNA in plasma in precancerous lesions and in the early stages of cancer is extremely insignificant, which, together with its invasiveness, in my opinion, does not allow it to be considered either as a screening method or as a key confirmatory method. In this regard, formulations like ‘in the CC context, the HPV-ctDNA isolation and detection could be coupled with the standard Pap smear test’ (line 324) and even ‘the use of non-invasive blood sample to use as LBs for isolating HPV-ctDNA would give the possibility to expand the screening also to developing countries’ (line 326) are at least questionable.

These remarks relate to the discussion on the diagnostic value of circulating cell-free DNA at early cervical cancer detection. Regarding the establishment of tumor burden, personalized monitoring of patient response to anticancer therapy and early relapse or recurrence prediction, the results obtained in recent years are indeed promising. In general, the manuscript is written in clear language, although it contains some easily removable typos like ‘proxstate’ (line 294) and stylistic errors.

The review can be published provided some revision of the discussion.

Comments on the Quality of English Language

The manuscript is written in clear language, although it contains some easily removable typos and stylistic errors.

Author Response

The article addresses an important topic concerning usability of circulating tumor DNA analysis in cervical cancer management. The amount of information in this area is growing rapidly, which makes the emergence of new reviews, involving the most recent data, certainly useful for the reader who wants to get more familiar with the topic.

However, I have a few comments on the conclusions that the authors make based on the reviewed data.

Answer: Thank you for your valuable comments. Indeed, they are really helpful at improving our manuscript and the general content of the text. Taking into consideration all of your suggestions we modified the manuscript according to each of your comments accordingly. Moreover, since we fully agree with your opinion that circulating DNA in plasma cannot always be detected, thus HPV-ctDNA is not yet a valid screening or key confirmatory method, we change and modify the suggested parts in a future-depented manner when enough evidences are available so as this method can contribute significantly in CC early stages precancerous lessons with a possible accurate predictive value. Following your last suggestion we also checked the manuscript carefully for any type of errors either removable typos or stylistic

Please, see the new lines corresponding to each of your comments respectively:

Some phrases sound very strange for me, i.e. ‘The standard method utilized for CC screening in the global population is the histological Pap smear test that gives a response on later stages of this disease, albeit missing the modification on precancerous lesions or post-therapeutic changes and metastatization’ (line 18). Did the authors mean cytological Pap smear? Even if so, cytological Pap smear, however subjective it could be, proved to be effective at detecting cervical precancerous lesions.

Answer: Amended in new lines 17-20: “The standard method utilized for CC screening in the global population is the cytological Pap smear test that despite its effective validity on giving precancerous lesions and a response on layer stages of this disease, a better screening and diagnostic reability as well as an improvement on the specificity and sensitivity are needed.”

The statement ‘human papillomavirus (HPV) contributes in the 99% of the cervical cancer cases’ (line 36) is not supported by the large body of evidence. Even in the cited article of Kusakabe et al. it is stated that only approx. 95% of squamous cervical carcinomas are caused by HPV, and in cervical adenocarcinomas the proportion of HPV-negative cases is 10-15%.

Answer: Amended in new lines 35-37: “The oncogenesis of this type of gynecological cancer is mostly correlated with infection from human papillomavirus (HPV) with at least the 80% of cervical adenocarcinoma to be HPV associated”

The formulation ‘HPV detection has been routinely assessed by Pap smear during CC screening’  (line 46) looks confusing as the term ‘Pap’ relates to the mode of staining of the smear which is not used for HPV detection.

Answer: Amended in new lines 57-59: “Abnormal cells caused by HPV has been usually detected at the time of Pap smear during CC screening, even if this method is not applicable for potential recurrence estimation”

The definition of HPV-DNA test as “molecular” Pap test (line 48) is not common and, to my view, not correct. This term is much more convenient for the molecular gene expression profiling of the epithelial cells, reflecting the risk of cervical lesions, than for the HPV DNA testing.

Answer: Thank you for your comment. We understand your concern about the terminology, nevertheless we are utilizing the term as the authors reported in the literature cited at the end of the sentence. To clarify the difference between the so called “molecular” and classical cytological Pap test, we amended the text in new lines 59-60: “In addition, the so called “molecular” Pap test (HPV-DNA test), differently from the cytological one”

Without a doubt, multiple biomarkers of cervical lesions can be of some value in patient management and successfully complement each other. However, in my opinion, the authors, commenting on literature data, substantially exaggerate the potential for detecting circulating HPV DNA using liquid biopsy. The end of the statement ‘Alongside cervical cytology, the detection of serum HPV-DNA can be used to detect the presence of an oncological case also in negative cytology test results, making it the only tool for proper CC diagnosis’ (lines 94-96) definitely must be softened.

Answer: Amended in new lines 104-108: “ The landscape of biomolecules detectable in serum liquid biopsies (LB) is enriched by the molecular biomarkers, such as the high-risk Human papilloma virus (HPV) DNA [41]. Alongside cervical cytology, the detection of serum HPV-DNA can be used in CC diagnosis since it can detect the presence of an oncological case also in negative cytology test results [42,43].”

The situation when as much as ‘4% of cfDNA in plasma is derived from cancer cells’ (lines 102-103) is not universal in cervical cancers, and even with invasive squamous cell carcinoma, and even more so with HSIL, HPV DNA cannot always be detected in plasma, that is supported by multiple publications in the field. In this regard, the negative predictive value of detecting circulating DNA in plasma in precancerous lesions and in the early stages of cancer is extremely insignificant, which, together with its invasiveness, in my opinion, does not allow it to be considered either as a screening method or as a key confirmatory method. In this regard, formulations like ‘in the CC context, the HPV-ctDNA isolation and detection could be coupled with the standard Pap smear test’ (line 324) and even ‘the use of non-invasive blood sample to use as LBs for isolating HPV-ctDNA would give the possibility to expand the screening also to developing countries’ (line 326) are at least questionable.

Answer: Amended in new lines 119-121: “On average, 6 ng of cfDNA can be isolated for each mL of plasma, of which a very low percentage or not at all is derived from cancer cells”

And lines 351-359: “In the CC context, the HPV-ctDNA isolation and detection could be coupled with the standard Pap smear test when a large amount of evidence exists and may support its accuracy and validity, giving a chance to provide insights on the lesion stage and a better follow up. Furthermore, the probable use of non-invasive blood sample to use as LBs for isolating HPV-ctDNA may offer a possibility to expand the screening also to developing countries, even though coming with some limitations. As mentioned also before, ear-ly-stage CC could result as a false-negative due to the low fractions of viral ctDNA [80,83], while this method would be extremely helpful in detecting locally advanced CC [80,82,83].”

These remarks relate to the discussion on the diagnostic value of circulating cell-free DNA at early cervical cancer detection. Regarding the establishment of tumor burden, personalized monitoring of patient response to anticancer therapy and early relapse or recurrence prediction, the results obtained in recent years are indeed promising. In general, the manuscript is written in clear language, although it contains some easily removable typos like ‘proxstate’ (line 294) and stylistic errors.

The review can be published provided some revision of the discussion.